# Composition of the Gut Microbiota Associated with the Response to Immunotherapy in Advanced Cancer Patients: A Chinese Real-World Pilot Study

**DOI:** 10.3390/jcm11185479

**Published:** 2022-09-18

**Authors:** Xi Cheng, Jiawei Wang, Liu Gong, Yong Dong, Jiawei Shou, Hongming Pan, Zhaonan Yu, Yong Fang

**Affiliations:** 1Department of Medical Oncology, Sir Run Run Shaw Hospital, Zhejiang University School of Medicine, 3 East Qingchun Road, Hangzhou 310016, China; 2Hangzhou D.A. Medical Laboratory, Hangzhou 310030, China

**Keywords:** 16S rRNA, cancer, gut microbiota, immunotherapy, microbial network

## Abstract

Background: The composition of the gut microbiota is associated with the response to immunotherapy for different cancers. However, the majority of previous studies have focused on a single cancer and a single immune checkpoint inhibitor. Here, we investigated the relationship between the gut microbiota and the clinical response to anti-programmed cell death protein 1 (PD-1) immunotherapy in patients with advanced cancers. Method: In this comprehensive study, 16S rRNA sequencing was performed on the gut microbiota of pre-immunotherapy and post-immunotherapy, of 72 advanced cancer patients in China. Results: At the phylum level, *Firmicutes*, *Bacteroidetes*, *Proteobacteria*, and *Actinobacteria* were the main components of the microbiota in the 72 advanced cancer patients. At the genus level, *Bacteroides* and *Prevotella* were the dominant microbiota among these 72 patients. The PD_whole_tree, Chao1, Observed_species and Shannon indices of R.0 and R.T were higher than those of NR.0 and NR.T. The results of LEfSe showed that *Archaea*, *Lentisphaerae*, *Victivallaceae*, *Victivallales*, *Lentisphaeria*, *Methanobacteriaceae*, *Methanobacteria*, *Euryarchaeota*, *Methanobrevibacter*, and *Methanobacteriales* were significantly enriched in the response group before immunotherapy (R.0), and the *Clostridiaceae* was significantly enriched in the non-response group before immunotherapy (NR.0) (*p* < 0.05). *Lachnospiraceae* and *Thermus* were significantly enriched in the response group after immunotherapy (R.T), and *Leuconostoc* was significantly enriched in R.0 (*p* < 0.05). ROC analysis showed that the microbiota of R.T (AUC = 0.70) had obvious diagnostic value in differentiating Chinese cancer patients based on their response to immunotherapy. Conclusions: We demonstrated that the gut microbiota was associated with the clinical response to anti-PD-1 immunotherapy in cancer patients. Taxonomic signatures enriched in responders were effective biomarkers to predict the clinical response. Our findings provide a new strategy to improve the efficiency of responses to immunotherapy among cancer patients.

## 1. Introduction

Immunotherapy has become a major therapeutic strategy in oncology. The recent clinical success achieved with immune checkpoint inhibitors, for example, by using blocking antibodies against programmed cell death ligand 1 (PD-L1), programmed death-1 (PD-1), and cytotoxic T lymphocyte antigen-4 (CTLA-4), illustrates the potential of the immune system in the elimination of cancer cells. A variety of immune checkpoint inhibitors have been approved by the U.S. Food and Drug Administration (FDA) for clinical use in multiple types of cancer. However, the current effectiveness of cancer immunotherapy is not satisfactory in all patients. Even if the initial treatment is effective, some patients still undergo early drug resistance to immune checkpoint inhibitors. Patient intrinsic factors, tumor stromal intrinsic factors and environmental factors (such as the gut microbiome) might contribute to the failure of immune checkpoint blockade [1]. Therefore, the identification of reliable biomarkers to ensure the optimization of immunotherapy would be of paramount interest. It has been reported that the effects of the specific gut microbiome can help to identify the microbe–host interaction networks that shape the host’s immune system, with the goal of manipulating these interactions for host’s health [2]. The gut microbiome may influence anti-tumor immune responses via innate and adaptive immunity, and therefore may improve the therapeutic responses.

The 16S ribosomal RNA (rRNA)-based sequencing of gene amplicons and the deoxyribonucleic acid (DNA) sequencing of patient fecal samples could identify subsets of the microbiome. For example, an increase in the permeability of the upper gastrointestinal tract could lead to the translocation of *Enterococcus hirae* from the small intestine to the spleen, as well as the accumulation of *Barnesiella intestinihominis* in the colon, which causes a coordinated immune-stimulatory effect on antitumor immune system responses [3]. The oral administration of *Bifidobacterium* could augment the dendritic cell function, leading to enhanced CD8(+) T cell priming and accumulation in the tumor microenvironment and mediating the response to immunotherapy [4].

The role of the gut microbiota in modulating cancer responses to immunotherapy has received increased attention in recent years. For metastatic melanoma patients receiving anti-PD-1-based immunotherapy, gut bacterial species more abundant in responders include *Bifidobacterium longum*, *Collinsella aerofaciens*, and *Enterococcus faecium* [5]. Furthermore, in melanoma patients undergoing anti-PD-1-based immunotherapy, significant differences were observed in the diversity and composition of the gut microbiome of responders versus non-responders, with significantly higher α−diversity and relative abundance of bacteria of the *Ruminococcaceae* family in responders [6]. However, the above-mentioned studies were all well designed clinical trials, and all enrolled patients met strict screening requirements. It should also be highlighted that variations exist between the different studies, which included patients with distinct genetic patterns and dietary habits, and clinical trials that were conducted in different geographic locations within the United States or Europe.

In clinical practice, there are obvious differences in the dietary structure of China and other countries. Chinese patients often have different physical conditions, and there are also many tumor patients over 75 years old. Since tumors are systemic diseases, patients are often accompanied by various complications. Faced with such a complex population, whether similar results are obtained between cases needs to be further confirmed by a real-world study. Additionally, a real-world study is expected to enable the development of more effective combination therapy strategies for immune checkpoint inhibitors and the advancement of precision medicine strategies.

Previous research to date has focused more on a single tumor type such as melanoma or lung cancer, rather than the diversity in real-world China. Based on previous results, the main objectives of the current study were to describe the association between immunotherapy clinical response and gut microbiota in different Chinese cancer patients, and to explore whether there are taxonomic signatures of gut microbiota that can predict clinical response or act as a biomarker.

## 2. Materials and Methods

### 2.1. Patients and Medications

This study is a single-center, observational, prospective study. Cancer patients treated with immunotherapy were prospectively enrolled at the Department of Medical Oncology, Sir Run Run Shaw Hospital, Zhejiang University between September 2019 and April 2020. This study was performed with the approval of the Ethics Committee of Sir Run Run Shaw Hospital, Zhejiang University. The enrollment criteria were as follows: (1) patients aged 18 years and above with adequate organ functions (e.g., neutrophil count ≥ 1.5 × 10^9^/L. platelet count ≥ 100 × 10^9^/L, hemoglobin ≥ 80 g/L, serum bilirubin ≤ 1.5 × upper normal limit, transaminase ≤ 3 upper normal limit, calculated creatinine clearance ≥ 60 mL/min); (2) histologically confirmed cancer, regardless of the tissue of origin; (3) had not received previous immunotherapy; (4) measurable disease by Response Evaluation Criteria in Solid Tumors, version 1.1 (RECIST version 1.1); (5) no synchronous or metachronous cancer; (6) had not received antibiotic treatment in the preceding 2 months. The patients with actively progressing brain metastases or a history of serious autoimmune disease were excluded from our study. All patients were enrolled after signing an informed consent. All procedures were carried out in accordance with the Declaration of Helsinki. All patients underwent a pre-treatment clinical workup.

The feces were collected at baseline (V0) and 6 weeks after the initiation of treatment (VT). The choice of immunotherapy was at the discretion of the treating oncologist; typically, Nivolumab was given every 2 weeks, and Pembrolizumab, Sintilimab, Camrelizumab and Toripalimab were given every 3 weeks. All patients were followed up with a physical examination, serum tumor marker evaluation, chest computed tomography (CT) and abdominal CT after every two cycles of immunotherapy. When necessary, a whole-body bone scan, positron emission tomography/computed tomography (PET/CT) scan, and cranial and abdominal magnetic resonance imaging (MRI) were additionally performed. After the completion of all cycles of immunotherapy, each patient was monitored every 3 months until the confirmation of disease progression.

In this study, we regarded the response to immunotherapy at 6 months after the initiation of treatment as the primary endpoint. Eligible patients were classified as responders (R) or non-responders (NR) based on radiographic assessment using the Response Evaluation Criteria in Solid Tumors (RECIST 1.1) criteria at 6 months after the initiation of treatment. Patients with good immunotherapy-related responses were defined as responders with long-term benefits, i.e., with an objective response (complete or partial response or stable disease lasting at least 6 months). Patients with progression or stable disease lasting less than 6 months were defined as non-responders.

Fecal samples were collected at baseline (V0) from each patient before the first administration of immunotherapy and 6 weeks after the initiation of treatment (VT). The fecal samples were stored at −20 °C until analysis. The samples represented four groups: response group before immunotherapy (R.0), response group after immunotherapy (R.T), non-response group before immunotherapy (NR.0) and non-response group after immunotherapy (NR.T).

### 2.2. DNA Extraction

In 3 days after collection, total DNA was extracted from fecal sample aliquots (150 mg) using the PowerMax DNA isolation kit (MoBio Laboratories, Carlsbad, CA, USA), following the manufacturer’s instructions. The tissue lyser (YMY-200, Saiyasi) was used to facilitate DNA extraction. The quantity and quality of extracted DNA were measured using a NanoDrop ND-1000 spectrophotometer (Thermo Fisher Scientific, Waltham, MA, USA) and agarose gel electrophoresis, respectively.

### 2.3. The 16S rRNA Amplicon Pyrosequencing

The extracted DNA was subjected to polymerase chain reaction (PCR) to amplify the V4 region of bacterial 16S rRNA, using the forward primer 515F (5′-GTGCCAGCMGCCGCGGTAA-3′) and the reverse primer 806R (5′-GGACTACHVGGGTWTCTAAT-3′). Sample-specific paired-end 6 bp barcodes were incorporated into the TrueSeq adaptors using PCR, which was performed prior to multiplex sequencing. PCR amplicons were purified with Agencourt AMPure XP Beads (Beckman Coulter, Indianapolis, IN, USA) and quantified using the Qubit dsDNA HS Assay Kit. After quantification, amplicons were pooled in equal amounts, and paired-end 2 × 150 bp sequencing was performed using the Illumina NovaSeq6000 platform at the Precision Diagnosis Center, Dian Diagnostics Group Co., Ltd. (Hangzhou, China).

### 2.4. Sequence Analysis

The Quantitative Insights into Microbial Ecology (QIIME, v1.9.0) pipeline was employed to process the sequencing data. Briefly, raw sequencing reads were assigned to each sample according to the paired sample barcodes. The low quality reads were filtered. Paired-end reads were assembled using Vsearch V2.4.4 and operational taxonomic units (OTUs) were identified using Vsearch V2.4.4, as well as dereplication, clustering, and detection of chimeras. OTU taxonomic classification was conducted by Vsearch by comparing the representative sequences set against the Greengeen database. An OTU table was generated to record the abundance of each OTU in each sample and the taxonomy of these OTUs.

### 2.5. Bioinformatics and Statistical Analysis

Sequence data analyses were mainly performed using QIIME and R packages (v3.2.0). OTU-level alpha diversity indices, such as the Chao1 richness estimator, ACE metric (Abundance-based Coverage Estimator), PD_whole_tree, Shannon diversity index and Simpson index, were calculated using the OTU table in QIIME. Abundance curves depicting ranked OTU levels were generated to compare the richness and evenness of OTUs among samples. Beta diversity analysis was performed to investigate the structural variation of microbial communities across samples using UniFrac distance metrics and the data were visualized via principal coordinate analysis (PCoA). Differences in the UniFrac distances for pairwise comparisons among groups were determined using the Student’s *t*-test and the Monte Carlo permutation test with 1000 permutations, and were visualized through box-and-whisker plots. The significance of differentiation of microbiota structure among groups was assessed by PERMANOVA (permutational multivariate analysis of variance) using the R package “vegan”. A Venn diagram was generated to visualize the shared and unique OTUs among samples or groups using the R package “VennDiagram”, based on the occurrence of OTUs across samples/groups regardless of their relative abundance. Taxa abundances at the phylum, class, order, family, genus and species levels were statistically compared among samples or groups by the Kruskal test from the R stats package. LEfSe (Linear discriminant analysis effect size) was performed to detect differentially abundant taxa across groups using the default parameters. Random forest analysis was applied to discriminate the samples from different groups using the R package “randomForest” with 1000 trees and using all default settings. The expected “baseline” error was also included, which was obtained by a classifier that simply predicts the most common category label. Microbial functions were predicted by PICRUSt (phylogenetic investigation of communities by reconstruction of unobserved states) and BugBase.

### 2.6. Statistical Analyses

Categorical baseline variables were compared using the Fisher’s exact test or the Chi-squared test and continuous baseline variables were compared using the *t*-test. Associations between microbiota dominant profiles and immunological parameters were assessed with the Spearman’s correlation coefficient and a two-sided Wilcoxon test. Analyses were performed with SPSS software (version 23.0). A *p* < 0.05 was considered statistically significant. No adjustment for multiple comparisons was made because of the exploratory component of the analyses.

## 3. Results

### 3.1. Characteristics of the Study Population

A total of 144 fecal samples were submitted for 16S rRNA gene sequencing from 72 patients with advanced cancer (III–IV stage) before and after receiving immunotherapy. The clinical characteristics of patients are listed in Table 1. The median age of patients at diagnosis was 63 years (range 29–81 years). Of the patients, 51 (70.83%) were male and 21 (29.17%) were female. There were 18 cases (25%) diagnosed as non-squamous non-small cell lung cancer, 14 cases (19.44%) of lung squamous cell carcinoma, 7 cases (9.72%) of hepatocellular carcinoma, 5 cases (6.94%) of gastric cancer, 5 cases (6.94%) of colorectal carcinoma, 5 cases (6.94%) of melanoma, 4 cases (5.56%) of nasopharyngeal carcinoma, 3 cases (4.17%) of cervical cancer, 2 cases (2.78%) of small-cell lung cancer, and other cancers (1 case of laryngeal cancer, 1 case of osteosarcoma, 1 case of renal pelvic carcinoma, 1 case of bladder cancer, 1 case of pancreatic cancer, 1 case of esophageal cancer, 1 case of ureteral cancer, 1 case of mediastinal carcinoma, and 1 case of cholangiocarcinoma). As for therapy, 45 cases (62.5%) had received chemotherapy, 28 cases (38.89%) had received radiotherapy, 41 cases (56.94%) had undergone surgery, and 26 cases (36.11%) had received targeted therapy prior to immunotherapy. In addition, 56 cases (77.78%) underwent immunotherapy combined with chemotherapy. All patients had an ECOG PS of zero or one.

We set the response to immunotherapy at 6 months after the initiation of treatment as the cut-off valve. The patients were classified as responders (*n* = 33) group or non-responders (*n* = 39). We compared the baseline characteristics between the two groups and no significant differences were observed. The characteristics of the responders and non-responders, and the results of statistical analysis, are summarized in Table 2. All *p* values were >0.05, indicating no significant differences between the two groups.

### 3.2. Taxonomic Profiles of Cancer Patients Pre- and Post-Immunotherapy

From all 144 fecal samples, 123,970 raw reads were generated and filtered to 115,656 high quality reads of the 16S rRNA gene (V4) region, per sample. The datasets were classified into 9708 OTUs, with 5393 OTUs being common, 7900 OTUs with R.0, 7662 OTUs with R.T, 7578 OTUs with NR.0, and 7982 OTUs with NR.T (Figure 1A). At the phylum level, *Firmicutes*, *Bacteroidetes*, *Proteobacteria*, and *Actinobacteria* were the main components of the microbiota in the four groups, accounting for 98% of the total (Figure 1B). *F_Bacteroidaceae* (22.55–25.26%), *f_Lachnospiraceae* (12.51–15.68%) and *f_Ruminococcaceae* (11.89–16.01%) were the dominating microbiota in the four groups. In individual groups, *f_Prevotellaceae* (10.90–12.03%) was abundant in R.0 and R.T, *f_Veillonellaceae* (9.65%) was abundant in NR.0, and *f_Enterobacteriaceae* (11.50%) was abundant in NR.T. At the genus level, *g_Bacteroides* (22.55–25.26%) and *g_Prevotella* (7.44–12.03%) were the dominating microbiota in the four groups. In the individual groups, *g_Faecalibacterium* (5.01%) and *g_Megamonas* (3.46%) were abundant in R.0, *g_Faecalibacterium* (4.29%) and *g_Bifidobacterium* (2.66%) were abundant in R.T, *g_Faecalibacterium* (4.78%) and *g_Bifidobacterium* (3.67%) were abundant in NR.0, and *g_Veillonella* (3.43%) and *g_Lachnospira* (2.98%) in were abundant NR.T (Figure 1C). The dominating bacteria at the family level were not determined, because of failing classification at the genus level.

### 3.3. Overall Microbial Richness and Diversity of Cancer Patients

We found that neither α-diversity nor β-diversity was significantly different the between four groups. However, there was an obvious increase in the PD_whole_tree, Chao1, Observed_species and Shannon index of R.0 and R.T compared with NR.0 and NR.T. Interestingly, the levels of these four indices were elevated after immunotherapy (Table 3).

In the PCoA results, different colored dots represent different groups. PCoA could not completely differentiate between the four groups of patients (PC1 = 37.58%, PC2 = 11.84%, Figure 2).

### 3.4. Significantly Enriched Microbiome and Functional Pathways in the Responder Group

Before immunotherapy, the results of LEfSe showed that the Archaea, Lentisphaerae, Victivallaceae, Victivallales, Lentisphaeria, Methanobacteriaceae, Methanobacteria, Euryarchaeota, Methanobrevibacter, and Methanobacteriales were significantly enriched in R.0, and Clostridiaceae was significantly enriched in NR.0 (*p* < 0.05, Figure 3A). The responders were compared before and after immunotherapy; Lachnospiraceae, Thermus were significantly enriched in R.T, and Leuconostoc was significantly enriched in R.0 (*p* < 0.05, Figure 3B). The function prediction of Picrust showed that L3_Stilbenoid diarylheptanoid and gingerol biosynthesis were significantly enriched in R.0 compared with NR.0 (*p* < 0.05, Figure 3C). When the responders were compared before and after immunotherapy, L3_Glycosaminoglycan degradation was significantly enriched in R.0 (*p* < 0.05, Figure 3D). ROC analysis showed that the microbiota (R.0, the area under the receiver operating characteristic curve (AUC) = 0.65; R.T, AUC = 0.70; NR.0, AUC = 0.62; NR.T, AUC = 0.64) had obvious diagnostic value in differentiating the four groups of cancer patients (Figure 3E).

### 3.5. The Microbial Network of the Gut Microbiome

Based on our data, 16 phyla were used for the network analysis, among which 20 showed a positive correlation and 16 showed a negative correlation. *Acidobacteria* was most positively correlated with TM7, and the correlation coefficient was 0.98. *Bacteroidetes* showed the strongest negative correlation with *Proteobacteria*, with a correlation coefficient of −0.57. Considering the main composition of the gut microbiota, *Bacteroidetes* was negatively correlated with *Firmicutes* (correlation coefficient = −0.49), *Proteobacteria* (correlation coefficient = −0.57), and *Actinobacteria* (correlation coefficient = −0.57). *Proteobacteria* was positively correlated with *Actinobacteria* (correlation coefficient = 0.16) (Figure 4).

## 4. Discussion

Multiple studies have highlighted the role of the gut microbiome in modulating immunotherapy efficacy in epithelial tumors, such as melanoma [5,7,8]. A previous study reported that the gut microbiome could impact on CpG–oligonucleotide immunotherapy responses, which in turn activated innate immune cells through TLR9 [9]. Investigators then focused on the role of the gut microbiome in shaping the T helper cell profile. The generation of specific subsets of Th17 and memory Th1 cells could modulate immunotherapy efficacy. Vetizou and colleagues showed that the efficacy of anti-CTLA-4 therapy was dependent on *B. fragilis*, *B. thetaiotaomicron* and *Burkholderiales* populations, with T cell responses specific for *B. fragilis* and *B. thetaiotamicron* being associated with immunotherapeutic efficacy. In addition, the reintroduction of *B. fragilis* cells or the adoptive transfer of *B. fragilis*-specific T cells could restore the efficacy of immunotherapy and reduce immune-mediated colitis through activation of Th1 cells. The cross-reactivity to bacterial antigens and tumor neo-antigens could activate the Th1 cells [10]. In terms of PD-1 and PD-L1 inhibitors, differences in responses have been linked to the gut microbiome composition. In particular, an increased abundance of *A. muciniphila* and *Enterococcus hirae* has been associated with anti-PD-1 inhibitor immunotherapy responders when compared with non-responders [8]. These responder and non-responder phenotypes have also been shown to be transmissible, as mice receiving a fecal microbiome transplants subsequently acquire donor responder or non-responder efficacy. The non-responsive phenotype can be rescued by the addition of *A. muciniphila* alone or in combination with *E. hirae*.

The findings from our study of the association between the gut microbiome and the immunotherapy response in advanced cancer patients are consistent with the results of previous studies [11,12,13]. Several studies examining the relationship between the diversity and composition of the gut microbiome and the clinical response during immunotherapy have identified that dysbiosis of the gut microbiome was related to adverse events and the clinical response to cancer immunotherapy [14]. These studies suggested that the gut microbiome prior to cancer treatment could be used as a predictor of the clinical response and recommended that assessment of the gut microbiome in cancer immunotherapy could improve patient care [15,16]. Extensive research revealed the synergistic activity of bacteria genera including *A. muciniphila*, *Alistipes indistinctus*, *Bacteroides*, *Bifidobacterium*, *Burkholderia cepacia*, *Collinsella aerofaciens*, and *Enterococcus*, as well as *Faecalibacterium* and *Gemmiger formicilis,* in immunotherapy. However, *Blautia obeum*, *Roseburia intestinalis*, and some combination of antibiotics compromised the efficacy of immunotherapy [8,17,18]. Similarly, a number of clinical studies have demonstrated a direct link between dysbiosis of the gut microbiome and cancer pathogenesis [19,20,21]. It is essential to develop a potential predictive biomarker; however, variations exist in the composition of the microbiome between studies, suggesting that there may be other unidentified factors that affect microbial diversity. Perhaps, a combination of commensal microbiome structure, tumor genomics, germline genetics, and other elements in a multi-parameter model may predict the clinical response to immunotherapy. Many clinicians and researchers rightly point out that further well-designed randomized controlled trials are required to explore the causal effects of the gut microbiome in cancer immunotherapy.

Lung cancer accounted for 47.2% of the total patients in our study. In a previous study, the diversity and stability of the gut microbiome appeared to be a biological marker for sensitivity to immunotherapy in lung cancer patients, and some specific species were found to predict the immunotherapy response. In the Checkmate 078 and Checkmate 870 studies of 37 patients with advanced NSCLC treated with nivolumab, investigators found that patients who responded to nivolumab had higher gut microbiome diversity at the beginning of treatment and a more stable gut microbiome composition during treatment. The enrichment of *Bifidobacterium longum*, *Alistipes putredinis* and *Prevotella copri* was associated with improved immunotherapy efficacy [22]. In addition, Song and colleagues found that higher β-diversity in the gut microbiome of lung cancer patients treated with immunotherapy predicted longer progression-free survival (PFS), and *Parabacteroides* and *Methanobrevibacter* predicted better cancer control [23]. Another study found that germ-free or antibiotic-treated mice transplanted with the gut microbiome from immunotherapy responders showed higher antitumor activities to PD-1 inhibitors than mice transplanted with non-responders’ feces. Sivan and coworkers found that mice administered *Bifidobacterium* presented with enhanced dendritic cell function and a concomitant intensified accumulation of CD8(+) T cells in the tumor beds, and combination treatment almost eliminated tumor outgrowth [4]. These results indicated that manipulation of the gut microbiome has the potential to enhance the efficacy of immunotherapy in lung cancer.

To examine the question of causal links between the gut microbiome and the response to immunotherapy, an additional review of the effects of gut microbiome modulation with fecal microbiome transplantation on immunotherapy was undertaken. Some studies investigated the effect of fecal microbiome transplantation in bacteria-depleted mice from melanoma immunotherapy responders, and the results indicated that the gut microbiome is a potential factor in modulating the effectiveness of immunotherapy. More recently, two clinical trials demonstrated that modulation of the gut microbiome with fecal microbiome transplantation from donors receiving anti-PD-1 inhibitor immunotherapy who showed a complete response into patients with refractory metastatic melanoma was safe and capable of enhancing the efficacy of cancer therapies [24,25]. Despite the differing compositions of the fecal microbiome transplants from the donors in these two studies, a subgroup of refractory metastatic melanoma patients (30% (3/10) and 40% (6/15)) demonstrated clinical responses in both studies [24,25]. Immunotherapy non-responders may not respond to fecal microbiome transplants for various reasons, including (i) an inability to respond to the tumor regardless of the microbiome composition because of the patient’s immune-deficient status or lack of tumor immunogenicity, (ii) an absence of taxa needed for immunotherapy effectiveness in the fecal microbiome transplant, or (iii) failure of the fecal microbiome transplant to successfully implant into the recipient.

Fecal microbiome transplant clinical trials have shown that this procedure is safe. Davar and colleagues, who used a single fecal microbiome transplant via colonoscopy at the beginning of the treatment protocol, reported good safety results with −72.9% of the immunotherapy-related adverse events (irAEs) being mild (grade I) and only three patients showing severe, grade III irAEs (two with fatigue, one with neuropathy) [25]. Another study that used colonoscopy at the beginning of the treatment protocol followed by repeated fecal microbiome transplant via stool capsules every 14 days reported no grade II or above irAEs, even in patients who developed grade III irAEs with previous immunotherapy [24,26]. These findings suggest that the combination of fecal microbiome transplants and immunotherapy is not only a more effective treatment but may also have a better safety profile. A treatment regimen with this type of duality, i.e., combining available Food and Drug Administration-approved, commonly used, oncological drugs with a highly available and easily re-produced organic compound (human feces), has sparked hope among both clinicians and cancer patients.

Several species associated with body mass index (BMI) have also been implicated in oncogenesis and immune dysregulation. *Fusobacteria*, a bacterial phylum with a role in oncogenesis, has been found to be elevated in the saliva and intestine of individuals with obesity [27,28]. Interestingly, *Fusobacteria* may activate NF-κB and other proinflammatory components, such as interleukin-1 (IL-1), interleukin-6 (IL-6), interleukin-8 (IL-8), tumor necrosis factor-α (TNF-α), matrix metalloproteinase 3 and cyclooxygenase-2, that are associated with intestinal carcinogenesis [29].

Immunotherapy has been a hot spot in tumor research. Real-world studies can provide a more comprehensive understanding of the clinical situation and beneficiary groups. The combination of immunotherapy and real-world study can provide more useful ideas and evidence for clinical treatment. At present, there are as many as 14 kinds of immune checkpoint inhibitors on the market in China. From the perspective of pharmacoeconomics, the formulation of immunotherapy treatment plans in China needs to take into account the patient’s work, living habits, economic level, and dietary habit, etc. Considering the complexity of immunotherapy regimens, a real-world study can more truly demonstrate the complex situation of the impact of gut microbiota on a patient’s response in actual clinical practice. Our study of a real-world patients with advanced various cancers who received immunotherapy treatment identified clinically relevant findings that may aid decision making: (1) Patients with non-responder gut microbiome phenotypes did not benefit from immunotherapy. So, the use of immunotherapy may increase these patients’ financial burden. (2) For those potentially at risk of early progression, we will explore the use of gut microbiome modulation to enhance the efficacy of immunotherapy in follow-up research.

Our study had several limitations. First, our study was conducted with a heterogeneous samples of cancer patients diagnosed with NSCLC, CRC, GC, melanoma, and HCC, among others. The composition of the human gut microbiome can be affected by various factors including diet, lifestyle, stress, environment and genetics, which not only complicates comparisons with animal model studies, but also needs to be controlled, as it may contribute toward population differences in identified differential microbial taxa. However, this heterogeneity may mean that the results are more applicable and reflective of the real-life cancer patient. Furthermore, our study analyzed fecal samples using a 16S rRNA sequencing method that can measure the composition of the gut microbiome from the phylum to genus level. To identify and validate a specific gut microbiome within a common microbiota community that contributes a direct link to immunotherapy responses in cancer patients, future international multi-center trials will be required to provide comprehensive and reliable data utilizing a standardized method of fecal sample analysis. Taking into account these limitations, studies with larger sample sizes are required to provide convincing evidence that can be implemented in clinical practice.

## 5. Conclusions

Our study demonstrated that the gut microbiota is associated with the clinical response to immunotherapy in various cancer patients in China. The gut microbiota of *Archaea*, *Lentisphaerae*, *Victivallaceae,* etc., as well as the function of L3_Stilbenoid diarylheptanoid, gingerol biosynthesis, and L3_Glycosaminoglycan degradation were significantly enriched in the response group before immunotherapy. These taxonomic signatures could act as effective biomarkers to predict the clinical response. Multilevel and multidimensional research designs integrating randomized controlled trials and real-world studies will become a part of personalized cancer therapy in the future.

## Figures and Tables

**Figure 1 jcm-11-05479-f001:**
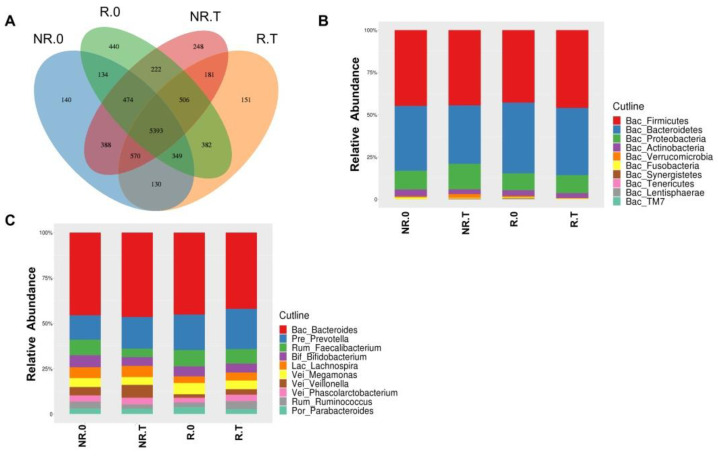
OTUs and microbial composition in four groups. (**A**) Venn diagram shows the number of OTUs in each group. Top 10 microbial composition at the phylum level (**B**) and genus level (**C**).

**Figure 2 jcm-11-05479-f002:**
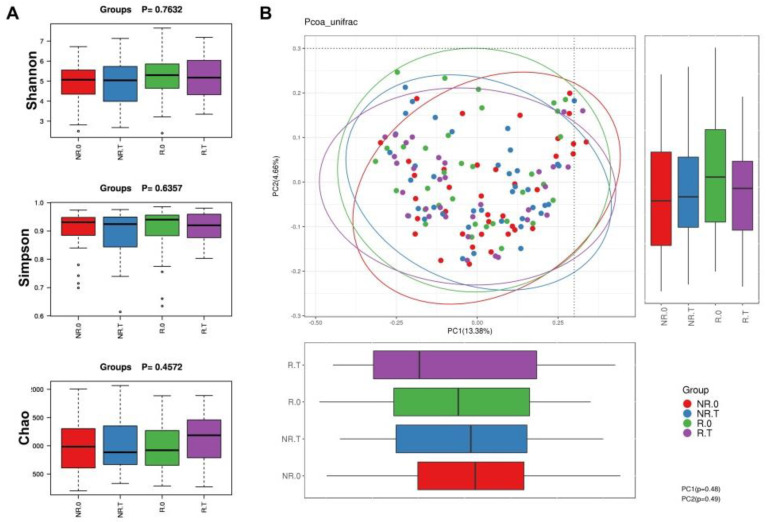
α−diversity index and PCoA results in four groups. (**A**) α−diversity index in each group. (**B**) PCoA with unweighted UniFrac distances.

**Figure 3 jcm-11-05479-f003:**
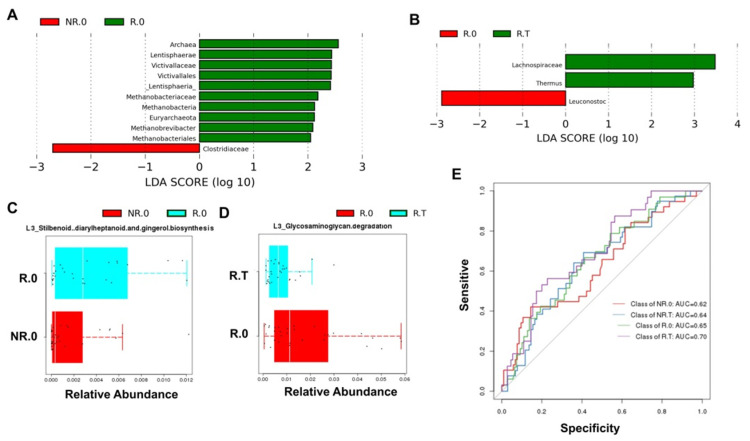
LEfSe analysis results and Picrust prediction in each group. (**A**) LEfSe analysis with significantly different representations in the R.0 group and NR.0 group. (**B**) LEfSe analysis with significantly different representations in the R.0 group and R.T group. (**C**) Picrust prediction of functional differences in the R.0 group and NR.0 group. (**D**) Picrust prediction of functional differences in the R.0 group and R.T group. (**E**) The AUC based on the microbiome for the R.0 group, R.T group, NR.0 group and NR.T group.

**Figure 4 jcm-11-05479-f004:**
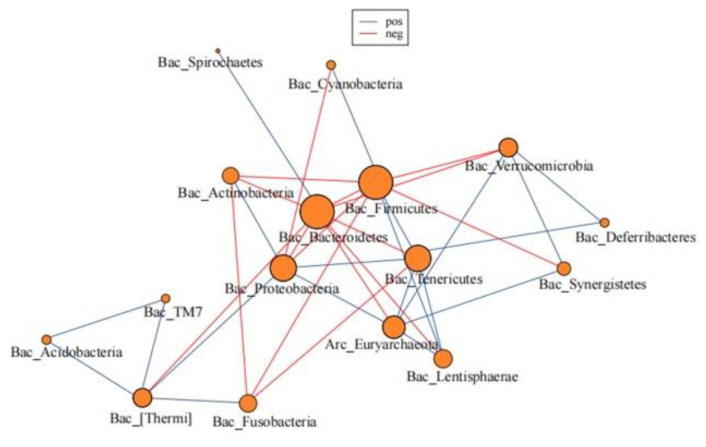
Microbial network of the gut microbiome. The red represents negative associations between two microbiomes, and green represents positive associations between two microbiomes. The number on the line represents the degree of correlation between two microbiomes.

**Table 1 jcm-11-05479-t001:** The characteristics of patients at baseline (*n* = 72).

Baseline Characteristics	*n* (%)
Age	
Median (range)	63 (29–81)
Gender	
Male	51 (70.83%)
Female	21 (29.17%)
Cancer	
Non-squamous non-small cell lung cancer	18 (25.00%)
Lung squamous cell carcinoma	14 (19.44%)
Hepatocellular carcinoma	7 (9.72%)
Gastric cancer	5 (6.94%)
Colorectal carcinoma	5 (6.94%)
Melanoma	5 (6.94%)
Nasopharyngeal carcinoma	4 (5.56%)
Cervical cancer	3 (4.17%)
Small-cell lung cancer	2 (2.78%)
Other cancer	9 (12.50%)
Immunotherapy monotherapy or combined with chemotherapy	
Immunotherapy monotherapy	16 (22.22%)
Immunotherapy combined with chemotherapy	56 (77.78%)
Received chemotherapy prior to immunotherapy	
Yes	45 (62.50%)
No	27 (37.50%)
Received radiotherapy prior to immunotherapy	
Yes	28 (38.89%)
No	44 (61.11%)
Underwent surgery prior to immunotherapy	
Yes	41 (56.94%)
No	31 (43.06%)
Received targeted-therapy prior to immunotherapy	
Yes	26 (36.11%)
No	46 (63.89%)

**Table 2 jcm-11-05479-t002:** The characteristics of responders and non-responders at baseline (*n* = 72).

	Responders (*n* = 33)	Non-Responders (*n* = 39)	*p* Value
Age			
Median (range)	66 (42–81)	61 (29–79)	
Gender			0.059
Male	27 (81.82%)	24 (61.54%)	
Female	6 (18.18%)	15 (38.46%)	
Cancer			0.639
Non-squamous non-small cell lung cancer	8 (24.24%)	10 (25.64%)	
Lung squamous cell carcinoma	6 (18.18%)	8 (20.51%)	
Hepatocellular carcinoma	2 (6.06%)	5 (12.82%)	
Gastric cancer	3 (9.09%)	2 (5.13%)	
Colorectal carcinoma	1 (3.03%)	4 (10.26%)	
Melanoma	2 (6.06%)	3 (7.69%)	
Nasopharyngeal carcinoma	3 (9.09%)	1 (2.56%)	
Cervical cancer	1 (3.03%)	2 (5.13%)	
Small-cell lung cancer	2 (6.06%)	0 (0.00%)	
Other cancer	5 (15.15%)	4 (10.26%)	
Immunotherapy monotherapy or combined with chemotherapy			0.850
Immunotherapy monotherapy	7 (21.21%)	9 (23.08%)	
Immunotherapy combined with chemotherapy	26 (78.79%)	30 (76.92%)	
Received chemotherapy prior to immunotherapy			0.077
Yes	17 (51.52%)	28 (71.79%)	
No	16 (48.48%)	11 (28.21%)	
Received radiotherapy prior to immunotherapy			0.169
Yes	10 (30.30%)	18 (46.15%)	
No	23 (69.70%)	21 (53.85%)	
Underwent surgery prior to immunotherapy			0.705
Yes	18 (54.55%)	23 (58.97%)	
No	15 (45.45%)	16 (41.03%)	
Received targeted-therapy prior to immunotherapy			0.054
Yes	8 (24.24%)	18 (46.15%)	
No	25 (75.76%)	21 (53.85%)	

**Table 3 jcm-11-05479-t003:** The indices of α-diversity analysis.

	PD_Whole_Tree	Chao1	Goods_Coverage	Observed_Species	Shannon	Simpson
R.T	20.07	677.15	0.94	464	4.89	0.90
R.0	18.77	594.96	0.94	414	4.95	0.90
NR.0	17.11	555.62	0.94	372	4.75	0.90
NR.T	19.19	627.71	0.95	433	4.75	0.88

## Data Availability

Not applicable.

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
