# Peer review of "Composition of the Gut Microbiota Associated with the Response to Immunotherapy in Advanced Cancer Patients: A Chinese Real-World Pilot Study"

_jcm, 2022, doi:10.3390/jcm11185479_

Round 1
Reviewer 1 Report
The aim of the study was a very interesting issue of an analysis of a gut microbiota composition with respect to the response to immunotherapy in advanced cancer patients. I suppose a lot of effort went into the study. The authors demonstrated that the gut microbiome is associated with the clinical response to anti-PD-1 immunotherapy in a quite diverse group of cancer patients and that the taxonomic signatures enriched in responders are effective biomarkers to predict the clinical response. Their findings provide a new strategy to improve the efficiency of a response to immunotherapy among cancer patients. Novelty and a significant impact of the results are really interesting.
However, my major concerns are:
· The credibility of the results would be higher for the study performed on a bigger group of patients with different types of cancers or, even better, one but more unified group of patients – focused on one particular cancer type or at least a group of cancers of the same origin. Therefore, I recommend an addition of “pilot study” into the title of the manuscript. Of course, if the authors are going to continue the research.
· There is no information on antibiotics application during patients’ treatment or at least at the time directly before the samples collection.
· Line 97 – units need correction.
· Line 126 – are these optimal conditions to preserve stool samples?
My minor concerns are:
· Keywords should be listed in alphabetical order, in my opinion.
· Was there any mechanical homogenization step during sample processing to facilitate DNA extraction?
· Tables 1 and 2 – suggest placing the data in consecutive order – starting each case from the most prevalent.
· Tables 1 and 2 – “Other cancer” definitely needs specification.
· Figures are mostly eligible.
· Lack of italics where it should be used.
All the points mentioned above do not decrease the total value of the research.
Reviewer 2 Report
· All abbreviations must be defined when used for the first time.
· All names of genus and species of microorganisms must be written in italics, e.g. lines 63-64.
· Give detailed aim of the study.
· Subtitles in Methods should be numbered.
· There are some typos in the whole text.
· How many patients did the experimental group have? How many test groups were there? What about a blind placebo group? It is not clearly presented in the methodology.
· How stool samples were collected, transported and stored until analysis?
· Authors should not use unused term “microflora” as there are no plants in the human body, but microbiota or microbiome in relation to human microorganisms.
· Figures and tables captions are not informative.
· Figures 3 and 4 are illegible.
· I miss the Conclusions section. What is the general conclusion of the research conducted?
